# An overview of the treatment interventions and assessment of fear-avoidance for chronic musculoskeletal pain in adults: A scoping review protocol

Sam Tan[1��], Anju Jaggi[2‡], Alex Tasker[3‡], Catherine Borra[4,5,6‡], Fraje Watson[7*��‡]

1 University College London, London, United Kingdom, 2 Royal National Orthopaedic Hospital, London, United Kingdom, 3 Bristol Veterinary School, University of Bristol, Bristol, United Kingdom, 4 Institute of Education, University College London, London, United Kingdom, 5 Trauma and Orthopaedics, Barts Health NHS Trust, London, United Kingdom, 6 Bone & Joint Health, Queen Mary University of London, London, United Kingdom, 7 Bioengineering, Imperial College London, London, United Kingdom

These authors contributed equally to this work.
‡ FW is the guarantor. ST and FW drafted the manuscript. AJ and FW conducted the initial Patient and Public Involvement activity which informed the current search strategy design. AT provided expertise in the data synthesis section. All authors read, provided feedback and approved the final manuscript.
* sam.tan.20@ucl.ac.uk

## Abstract

### Introduction

The Fear-Avoidance (FA) model aims to explain how an acute pain experience can develop into a persistent state. The FA model considers five core components: kinesiophobia, pain-related fear, catastrophisation, victimisation, and interpersonal social environment. Amongst these, kinesiophobia, tends to dominate the literature on chronic musculoskeletal pain. As a result, current reviews have not considered the other core components of the FA model when exploring its interventions. Moreover, several synonyms of the term kinesiophobia is not reflected in their search strategies. Coupled with the preference of particular study designs and outcome measures, this scoping review aims to provide and characterise an overview of treatment interventions that consider all study designs, relevant outcome measures, FA components, and FA component synonyms.

### Methods and analysis

Eligible studies will be in English or with an available English translation from 1970 onwards. Databases to be searched include Cochrane Central Register of Controlled Trials (CENTRAL), MEDLINE, Embase, The Allied and Complementary Database (AMED), PEDro, Web of Science, and grey literature. We will include studies involving participants ≥18 years old with chronic musculoskeletal pain, and interventions targeting FA and/or its components. Three review authors will independently screen

**Data availability statement:** No datasets were generated or analysed during the current study. All relevant data from this study will be made available upon study completion.

**Funding:** This study was funded by an internal award from UCL's Institute of Healthcare Engineering for Summer Studentships awarded to FW and AJ. The total award was £7030.00 covering 16-weeks full-time equivalent pay at the London Living Wage and £400 for project materials. The funder played no role in study design, data collection or analysis, decision to publish or preparation of the manuscript.

**Competing interests:** The authors have declared that no competing interests exist.

papers using preestablished eligibility criteria and conduct assessments of risk of bias, with a fourth independent researcher employed to resolve disagreements where found. Qualitative synthesis techniques will be used to characterise the interventions. Patient and Public Involvement (PPI) has been utilised to develop this protocol and will be conducted following completion of the systematic review to discuss and reflect on the findings.

## Ethics and dissemination

This systematic review does not require ethical approval as existing data will be used and the PPI to be conducted is an involvement activity rather than study data. The results will be disseminated through a peer-reviewed journal and via national and international conferences.

## Open Science Framework registration number

**This protocol is registered on Open Science Framework:** https://doi.org/10.17605/OSF.IO/NR37A

---

## Introduction

### Rationale

The Fear-Avoidance (FA) model was developed by Lethem et al in 1983 [1] to explain the pathways by which acute pain transitions to chronic using a biopsychosocial perspective [2,3]. According to the FA model, a self-reinforcing cycle of catastrophising, fear, hypervigilance, and further avoidance of movement can lead towards dysfunction, deconditioning, depression, and increased pain [4]. The FA model is made up of four main components: kinesiophobia, catastrophising, pain-related fear, and FA beliefs [5,6]. Some researchers have suggested adding victimisation, disability, self-efficacy, and input from the interpersonal social environment, suggesting that the model should extend beyond pain-related fear and highlights the significance of concurrent, and often competing, goals [3,7–9] (Fig 1).

In the assessment and treatment of chronic musculoskeletal pain, kinesiophobia may be described and understood by a range of synonyms (e.g., "fear of movement", "fear of (re)injury", "avoidance behaviour", "avoidance hypervigilance", "behavioural performance", "pain-related fear with impaired physical performance", and "avoidance of activity") which tend to dominate the research landscape [10–12]. Kinesiophobia is an important mediator of new and prolonged chronic pain and disability, and as such is an important element of the FA paradigm; rates are estimated at 51–72% of people with chronic pain [10,13]. Kinesiophobia presents a major challenge to successful physical rehabilitation; unchecked, it may contribute to the development of disuse syndrome, which is associated with additional physical and psychosocial impacts such as depression, muscle atrophy, and medication misuse [14]. These fundamental effects risk drawing practitioners focus away from less-reported aspects of the FA model.

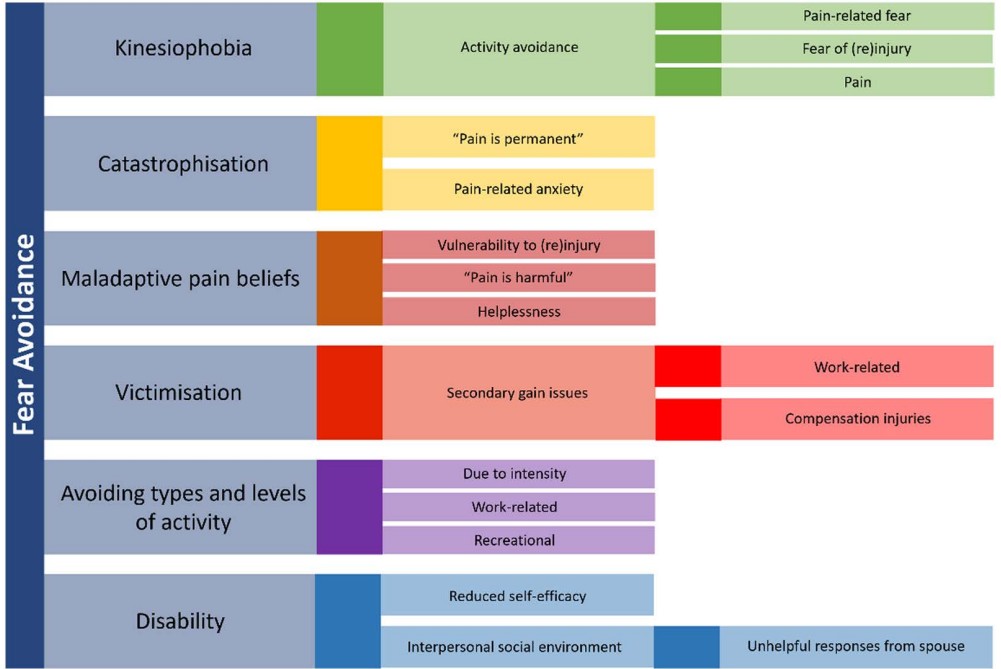

**Fig 1. The components of the Fear Avoidance model and potential implications.**

Several systematic reviews have focused on the results of interventions to treat kinesiophobia [10,15–18]; a focus which risks overlooking the multifactorial origins and drivers of chronic musculoskeletal pain. Bordeleau et al [10] recommends further research into the multi-faceted nature of kinesiophobia, encompassing physical, psychological, and social interventions. These concerns are further compounded by the nature and use of tools available to measure kinesiophobia; methods such as the Tampa Scale of Kinesiophobia (TSK) that is widespread and accessible [19]. Liu et al conducted a review of related measures and found that The Fear-Avoidance Beliefs Questionnaire (FABQ) and TSK were the most often used questionnaires [20,21]. Indeed, many systematic reviews have specified use of the TSK as part of their study inclusion criteria [10,16]. This further exacerbates the focus on kinesiophobia, and simultaneously dampens potential for assessment of the full FA model on chronic musculoskeletal pain. Moreover, current reviews have primarily included randomised controlled trials (RCT) in their search strategy, which restricts the diversity of interventions explored.

In addition to the TSK and FABQ, there are other FA model-adjacent measures including the Kinesiophobia Causes Scale (KCS) [22], Athlete Fear-avoidance Questionnaire (AFAQ) [6], and the Fear-Avoidance Components Scale (FACS) [8]. Some researchers have started to use a combination of the aforementioned measures, but it is unknown whether this is appropriate given the potential for discordance between each measure and the FA model construct [10,23–25].

The current literature on assessing FA and its components, beyond just kinesiophobia, is lacking. There is a need for a scoping review to provide a comprehensive overview of the current interventions for FA, that includes intervention studies assessing and/or treating FA by appreciating all its components on chronic musculoskeletal pain, as it is more patient-tailored and clinically useful. As such, utilising this study's interpretation of the FA model will influence the search criteria to include all relevant study designs and the current measures of FA components.

Finally, the voice of community collaborators, such as patients, carers, and allied health professionals is vital in ensuring patient-centred conclusions and recommendations for assessment and interventions of FA in musculoskeletal health. Therefore, the objectives of this scoping review are as follows:

1. Identify and characterise interventions used to treat FA in people aged 18 and over with chronic musculoskeletal pain by considering all study designs, relevant measures, and FA synonyms in all applicable healthcare settings;

2. Identify and describe the most common FA model-related measures that assess treatment interventions in people with chronic musculoskeletal pain;

3. engage with relevant community collaborators, such as patients and musculoskeletal clinicians, to discuss the findings regarding the current understanding of FA concepts, generate new ideas for interventions, and to ensure further research to be clinically relevant.

## Methods

This protocol will follow the guidance of the Joanna Briggs Institute (JBI) manual for scoping reviews [26], and adapted items from both the Preferred Reporting Items for Systematic Reviews and Meta-Analysis Protocols (PRISMA-P) [27] and the Preferred Reporting Items for Systematic reviews and Meta-Analyses extension for Scoping Reviews (PRISMA-ScR) [28,29]. This study is registered with the Open Science Framework (OSF) and the search will be conducted between 1st April – 1st July.

### Patient and Public Involvement Statement (PPI)

A panel of patients with chronic pain who attended the Royal National Orthopaedic Hospital, UK attended an online focus group about chronic pain and were reimbursed for their time. Regarding kinesiophobia, patients spoke about the importance of early intervention, education, and rehabilitation from encouraging, knowledgeable, and positive health professionals. Within the discussion, participants mentioned other aspects of fear avoidance, such as catastrophisation, and were concerned that their fear of movement would never fully disappear but felt hopeful that it could be dampened with correct management. Participants all agreed that peer support was vital to their personal 'journey' in managing fear of movement. Moreover, a concern was raised about how a participant felt that they were unable to take part in research studies due to their comorbidities, and so consequently believed that their views and experiences were not reflected in the research.

The authors felt that the term 'kinesiophobia' was quite uncommon among allied health professionals, musculoskeletal researchers, and patients, and that research may exist on this topic without naming it in such a way, and consequently be missed by systematic review searches. To combat this, the aim was to reach multiple health professionals to consider their understanding of kinesiophobia so that they could offer the terminology they use with other clinicians or their patients. The authors circulated the following question on X (formerly Twitter; www.x.com),

*Calling ALL health professionals! What do you mean when you say \*kinesiophobia\*? Please comment below + include a description of your job role + share with your network.*

This resulted in 20,001 impressions, 727 engagements, 23 retweets and 37 comments. Direct responses came from 36 individuals, representing physiotherapists, musculoskeletal researchers, orthopaedic surgeons, pain management clinicians, chiropractors, and sports trainers. Examples of responses include,

*"…being scared to move too much for fear of triggering a flare up of pain."*

*"…rationale of irrational fear of causing or increasing tissue damage…"*

*"when an individual doesn't have the knowledge, skills and/or confidence to move their body…"*

*"I don't tend to use the word clinically however, in children+young[sic] people I would describe kinesiophobia as the loss of spontaneous, fluent and playful movement due to a memory or thought that links movement to a negative and feared consequence..."*

*"…no patient has ever used the term unless learnt from a clinician."*

*"…I don't like when it's describe[sic] as "irrational" fear it makes me feel like we're blaming the person."*

*"…delineating terms may help us work more effectively with patients…"*

These interactions subsequently informed the design of our search terms and inclusion/exclusion criteria.

In addition to this protocol being informed by this PPI, another PPI activity will be conducted that will discuss the results of this scoping review with a focus group including patients, carers, and health professionals. This can include their impression of the findings, the potential for impact on them, and to gauge how well their experience is represented by the findings and generate ideas for future research. The nature of this PPI is an involvement activity, where the opinions of community collaborators were collected on the topic of FA, and so ethical approval is not required.

### Eligibility criteria

The eligibility criteria will follow the elements of the "Population", "Concept", and "Context" framework as outlined in the JBI guide [26].

The types of participants include papers describing individuals of any sex who are aged ≥18-years old with chronic musculoskeletal pain.

According to the International Classification of Diseases 11th Revision (ICD-11) [30], chronic musculoskeletal pain is categorised as either primary (ICD-11 code MG30.02) or secondary (ICD-11 code MG30.3), which involve bone(s), joint(s), muscle(s), vertebral column, tendon(s) or related soft tissue(s). Both categories will be included.

Chronic post traumatic pain (ICD-11 code MG30.20) where the aetiology of pain is predominantly trauma-induced and musculoskeletal in nature, will also be included. Typical examples of chronic post traumatic pain include chronic pain after acute back injury, and whiplash injury. Chronic whiplash injury, a condition with grading that reflects neuropathic involvement, is a highly relevant condition [31] that will still be included according to the ICD-11 definition as its aetiology is highly likely to be post traumatic.

When musculoskeletal components are the primary drivers of the pain, chronic peripheral neuropathic pain (code MG30.51) will be included. For instance, diabetic peripheral neuropathy is primarily agreed that the aetiology is from endocrine dysfunction [32], which would not be included. On the other hand, peripheral neuropathy from nerve compression injuries will be included due to the musculoskeletal element to it [33].

Included studies can contain homogenous or heterogenous populations and individual participants can experience multiple comorbid conditions.

The concept of the criteria is to include any intervention (e.g., physical, psychiatric, educational, psychosocial, behavioural) with or without comparison to a control group and regardless of outcome, due to the exploratory nature of this review. The intervention must target FA and/or its components by the context of the study. The questionnaire used to assess FA and/or its components can either be primary or secondary outcome. The included questionnaires used to measure fear avoidance are outlined in the search strategy section and Table 1. No limits nor filters will be used to restrict the study designs included.

The context of the studies will consider all settings (e.g., primary care, secondary care, tertiary care), time frames, and treatment lengths. The FA model was initially outlined by Lethem et al [1] and studies that explored psychological factors in chronic pain were increasingly prevalent in the 1970s, so the date range for included reports will be from 1970 onwards.

### Exclusion criteria

Studies on children, solely healthy subjects, chronic secondary musculoskeletal pain due to disease of the nervous system (ICD-11 code MG30.32), chronic secondary visceral pain (ICD-11 code MG30.4), non-musculoskeletal traumatic

**Table 1. Example search of the MEDLINE database on Ovid.**

| |
|---|
| 1. pain/ or exp acutepain/ or exp arthralgia/ or exp back pain/ or exp chronic pain/ or exp musculoskeletal pain/ or exp neck pain/ or exp neuralgia/ or exp nociceptive pain/ or exp pain, intractable/ or exp pain, postoperative/ or exp pain, procedural/ or exp pain, referred/ or exp pelvic pain/ or exp Failed Back Surgery Syndrome/ |
| 2. physical suffer*.mp. |
| 3. exp Complex Regional Pain Syndromes/ |
| 4. exp musculoskeletal diseases/ or sciatica/ |
| 5. arthritis/ or exp arthritis, infectious/ or exp arthritis, psoriatic/ or exp arthritis, rheumatoid/ or exp gout/ or exp osteoarthritis/ or exp periarthritis/ or exp sacroiliitis/ or exp spondylarthritis/ |
| 6. exp Neuropathology/ |
| 7. exp neck injuries/ or whiplash injuries/ |
| 8. exp catastrophization/ or exp kinesiophobia/ |
| 9. catastrophi?*.mp. |
| 10. kinesiophobi*.mp. |
| 11. Avoidance Learning/ |
| 12. "fear of movement".mp. |
| 13. "fear AND movement".mp. |
| 14. "fear of injur*".mp. |
| 15. "fear of reinjur*".mp |
| 16. "fear of re-injur*".mp. |
| 17. (fear and injur*).mp. |
| 18. (fear and reinjur*).mp. |
| 19. (fear and re-injur*).mp. |
| 20. "fear avoidance*".mp. |
| 21. "behavioural performance".mp. |
| 22. "behavioral performance".mp. |
| 23. "pain-related fear*".mp. |
| 24. "avoidance behaviour*".mp. |
| 25. "avoidance behavior*".mp. |
| 26. (avoidance and physical exertion).mp. |
| 27. impaired physical performance.mp |
| 28. "Tampa scale of kinesiophobia".mp. |
| 29. Fear-Avoidance Beliefs Questionnaire.mp. |
| 30. "Athlete Fear-Avoidance Questionnaire".mp. |
| 31. "Fear-Avoidance Components Scale".mp. |
| 32. "Kinesiophobia Causes Scale".mp. |
| 33. "Pain Anxiety Symptoms Scale".mp. |
| 34. Patient-Reported Outcomes Measurement Information System Pain Interference.mp. |
| 35. Pain Catastrophi?ing Scale.mp. |
| 36. 1 or 2 or 3 or 4 or 5 or 6 or 7 |
| 37. 8 or 9 or 10 or 11 or 12 or 13 or 15 or 16 or 17 or 18 or 19 or 20 or 21 or 22 or 23 or 24 or 25 or 26 or 27 |
| 38. 28 or 29 or 30 or 31 or 32 or 33 or 34 or 35 |
| 39. 36 AND 37 AND 38 |

injuries (e.g., burn injuries), and postoperative populations (e.g., following total knee arthroplasty) will be excluded. Review articles and secondary analysis papers will be excluded.

The searches will have no restrictions on language to avoid bias, and an effort to translate studies in languages other than English will be made. If this is not feasible, then it will be noted that these studies will be marked as "studies awaiting classification" and be included in the flow chart described in the PRISMA-ScR [34].

### Information sources

Cochrane Central Register of Controlled Trials (CENTRAL), MEDLINE, Embase, The Allied and Complementary Medicine Database (AMED), PEDro, Web of Science, and grey literature (e.g., medRxiv, ISRCTN protocol registry) will be searched using database-specific syntax detailed below. Reference lists of resulting articles will be searched for articles previously missed. The final list of selected articles will be entered into Connected Papers (www.connectedpapers.com/), which will also search for associated papers that could have been missed.

### Search strategy

The search strategy is formulated with assistance from an expert librarian using the Peer Review of Electronic Search Strategies (PRESS) Evidence-Based Checklist [35]. The search strategy is divided into three themes: [1] musculoskeletal conditions, [2] FA components and their synonyms, and [3] relevant outcome measures. These themes will be combined with the Boolean operator "AND".

Ovid will be used for the AMED, Embase, and MEDLINE databases whilst CENTRAL and PEDro will be searched on their own respective search engines. Key search terms will be used for every database and logically matched with their respective subject headings or Medical Subject Headings (MeSH) term system. For instance, currently Ovid has a "Map Term to Subject Heading" option, whilst CENTRAL uses MeSH terms.

The key search terms for the search strategy are:

1. **Search 1 (musculoskeletal conditions)**: "pain", any MSK "-algia" conditions (e.g., causalgia, neuralgia, epicondyalgia, etc.), MSK pain, physical suffering, MSK pain (any body part, e.g., elbow, knee, neck, back, etc.), "phantom limb", "complex regional pain syndrome", "sciatica", "neuropath*", arthritis (all forms, e.g., osteoarthritis, rheumatoid arthritis, etc.), "sacroilitis", "Failed back surgery syndrome", "persistent spinal pain syndrome", "whiplash"

2. **Search 2 (FA components)**: kinesiophobi*, "fear of mov*", "fear of injur*", "fear of reinjur*", "fear of re-injur*", "fear" AND "movement", "fear" AND "injur*", "fear" AND "reinjur*", "fear" AND "re-injur*", "fear avoidance", "behavioural performance", "behavioral performance", "avoidance behaviour", "avoidance behavior", "pain-related fear", "avoidance of physical exertion", "avoidance hypervigilance", "impaired physical performance", "scared of movement", "mov* anxiety", "avoidance learning", "catastrophi?*", "maladaptive pain belief", "pain belief", "victimi?ation"

3. **Search 3 (relevant outcome measures)**: "Tampa Scale of Kinesiophobia", "Kinesiophobia Causes Scale", "Fear-Avoidance Beliefs Questionnaire", "Fear-Avoidance Components Scale", "Athlete Fear-Avoidance Questionnaire", "Pain Anxiety Symptoms Scale", "Pain Catastrophi?ing Scale", "Patient-Reported Outcomes Measurement Information System Pain Interference"

4. **Search 4:** Search 1 AND Search 2 AND Search 3

Due to the iterative nature of scoping reviews, and as the search process becomes more familiar, the search may have different key search terms. This may mean incorporating those that can be discovered or removing ones that will decrease the sensitivity [26].

Table 1 shows an example search of the MEDLINE database using the Ovid platform:

### Study records

**Data management.** Search results will be exported from the database and compiled in Rayyan (www.rayyan.ai/). Rayyan is an online tool that is designed for screening reports in systematic reviews and allows convenient collaboration amongst the researchers, such as with remote working. There is also an additional feature that allows the automatic detection and removal of duplicates, which will be done at the 95% confidence threshold within the Rayyan settings. The setting for blinding will be turned on so the reviewers will be blinded to each other's decisions.

**Selection process.** Using Rayyan, three researchers including clinicians and academics (ST, FW, AJ) will independently review the search results to deem whether studies meet predetermined inclusion/exclusion criteria using the title and abstract. The full text of remaining studies will be reviewed independently to determine final inclusion in the systematic review. Where disagreements occur, an impartial researcher (AT) will help reach resolution through group discussion. A flow chart will be produced showing papers excluded at each stage, along with the reason as described in the PRISMA-P [27].

**Data collection process.** A data collection form detailing all data items will be used to extract information for included papers by at least two independent researchers (ST, FW, AJ, CB). Similarly to the selection process, where disagreements occur, an impartial researcher (AT) will help reach resolution through group discussion.. Where data is missing, two attempts will be made to contact the corresponding author of the papers at two-week intervals, to retrieve the necessary information.

**Data items.** Study identifiers, study design, cohort size, cohort location, intervention, participant gender, participant age, cohort diagnosis/disorder, cohort symptom characteristics, cohort comorbid diagnoses, dosage of intervention (e.g., frequency, intensity, periodicity), outcome measure (e.g., TSK), treatment outcome, comparative group (where applicable), power (where applicable and achievable), effect size (where applicable and achievable), level of evidence for therapeutic studies [36].

### Outcomes and prioritisation

The primary outcome will be a list of the identified and characterised treatment interventions used to treat FA components and the questionnaires used to assess outcome. Secondary outcomes will be the opinion of community collaborators through PPI to provide reflections and develop a list of future research priorities.

### Risk of bias of individual studies

Critical appraisal of individual studies is often omitted in scoping reviews (Peters et al., 2022), however providing an overview of the risk of bias amongst the selected studies can assist with describing the current level of quality in the literature.

For RCTs, the Risk of Bias (ROB) 2.0 tool [37] will be used to assess the risk of bias. For non-randomised studies, the Risk of Bias In Non-randomised Studies – of Interventions, Version 2 (ROBINS-I V2) tool [38] will be used. Both tools utilise a fixed set of domains of bias on the design, conduct, and reporting, which will all be assessed for every study. Both tools utilise relevant cut-off points that categorises the risk of bias as either "low risk of bias", "some concerns", or "high risk of bias". These cut-off categories will be based on the answers from each domain assessment using their respective flowchart algorithms for both the ROB 2.0 and ROBINS-I V2.

Analysis of bias will be carried out for all included studies twice between two reviewers (ST, FW) independently. Where disagreements occur, discussion led by a mediator (AT) will aim to resolve differences and result in agreement.

### Data synthesis

In line with JBI best practice and definitions, scoping reviews may employ similar thematic analysis techniques and methodologies but are not normally conducted to the same depth as systematic reviews. We chose to perform a modified thematic analysis with a realist synthesis approach to characterise the identified interventions [39,40].

Thematic analysis is an established qualitative analytical tool designed to reveal and organise themes in a hierarchical manner through a structured approach and has been tested and validated in similar studies [41]. For instance, this method is an iterative process that would be used to identify the core characteristics to the nature, types, and approaches of the interventions and assign codes to them. This will be continually refined by reflecting and collecting these codes into indicative proto-thematic areas, then assembling proto-themes into a wider narrative structure to aid discussion.

The characterisations generated during thematic analysis will be influenced by the author's personal positioning and experiences as a musculoskeletal clinician (ST). The realist approach refers to considering the context, proposed mechanisms, and the outcomes of the intervention, which will facilitate the construction of these themes. The results will be tested and interpreted by key community collaborators in the PPI groups. This qualitative approach will enable the robust integration of the scoping review and PPI reflexive data to better appreciate multiple perspectives on interventions.

## Presentation of the results

Tabulation and visual displays of the results will be created for the various aspects of the study, such as the flow chart for the selection process, study characteristics, individual risk of bias assessments, list of identified interventions, characterisations of the identified FA interventions, and FA outcome measurement tools used. Descriptive summaries will be provided on the main features of these aspects, such as the types of interventions included, which characterisations were constructed, and the distribution of outcome tools used.

Due to the broad and iterative nature of scoping reviews [29], the data presentation may be further refined but will still align with the study's objectives.

## Discussion

This systematic review will provide and characterise a broad overview of interventions intended to treat FA and evaluate the questionnaires used to assess their outcomes. Until now, many systematic reviews have focused on kinesiophobia as a single aspect of the FA model, rather than a more holistic approach, as is proposed here. A broad perspective of treatment interventions for FA related to chronic pain in musculoskeletal conditions could educate medical professionals and highlight areas of further research. Understanding how the outcome of these studies are assessed using questionnaires will allow recommendations to be made for future research and a stronger inclusion of the whole FA model rather than purely kinesiophobia.

A potential limitation of this protocol is the inclusion of participants with comorbidities because FA often results from, or is commonly comorbid with, other diagnoses. Whilst this may increase the risk of selection bias, it can also ensure results are more generalisable [42], in keeping with our PPI focus group findings. Additionally, assessment of risk of bias are not usually done in scoping reviews, but we felt it would provide a more comprehensive perspective on the quality of the current literature. Furthermore, the iterative process of a scoping review may introduce the addition or omission of key search terms described in the protocol, however the three themes will be maintained.

In carrying out this work, we also hope to introduce an interesting PPI framework that could be used and expanded upon in future research. Public involvement could empower people with chronic musculoskeletal pain to take ownership of their treatment and seek out knowledgeable professionals who deliver evidence-based care. Additionally, it will ensure our research aims, perspectives and priorities are informed and directed by the exact population we aim to help.

## Strengths and limitations of this study

- This scoping review provides a meaningful and patient-centred search in the context of fear-avoidance (FA), as the search strategy purposely includes additional components of FA and its synonyms.

- The search strategy will include eight relevant measures and all study designs to provide a diverse overview of the current interventions for FA in chronic musculoskeletal conditions.

- Utilising a combination of established qualitative methods to identify and characterise interventions for fear avoidance and integrating them with the opinions of the PPI community collaborators to better appreciate multiple perspectives and inform further research.

- This review will include the assessment of risk of bias of individual studies, which is not usually conducted in scoping reviews. This will aid in summarising and mapping the current quality of research evidence.

- The inclusion of participants with comorbidities increases generalisability of results at the risk of introducing additional biases and confounders.

- The objectives of this scoping review are not applicable to conduct a meta-analysis, but instead inform future research by clarifying the key concepts of FA and its interventions.

## Supporting information

**S1 Checklist. PRISMA-P.**
(DOCX)

## Author contributions

**Conceptualization:** Anju Jaggi, Alex Tasker, Catherine Borra, Fraje Watson.

**Funding acquisition:** Anju Jaggi, Fraje Watson.

**Investigation:** Sam Tan, Alex Tasker, Fraje Watson.

**Methodology:** Sam Tan, Anju Jaggi, Fraje Watson.

**Project administration:** Sam Tan, Anju Jaggi, Alex Tasker, Catherine Borra, Fraje Watson.

**Resources:** Anju Jaggi, Alex Tasker, Catherine Borra, Fraje Watson.

**Supervision:** Anju Jaggi, Alex Tasker, Catherine Borra, Fraje Watson.

**Visualization:** Sam Tan, Fraje Watson.

**Writing – original draft:** Sam Tan.

**Writing – review & editing:** Anju Jaggi, Alex Tasker, Catherine Borra, Fraje Watson.

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
