## [Decision Letter · Decision Letter 0]

11 Nov 2024

Dear Dr. Watson,

Thank you for submitting your manuscript to PLOS ONE. After careful consideration, we feel that it has merit but does not fully meet PLOS ONE’s publication criteria as it currently stands. Therefore, we invite you to submit a revised version of the manuscript that addresses the points raised during the review process.

We look forward to receiving your revised manuscript.

Kind regards,

André Pontes-Silva

Academic Editor

PLOS ONE

Journal Requirements:

Reviewers' comments:

Reviewer's Responses to Questions

**Comments to the Author**

1. Does the manuscript provide a valid rationale for the proposed study, with clearly identified and justified research questions?

Reviewer #1: Yes

Reviewer #2: Yes

2. Is the protocol technically sound and planned in a manner that will lead to a meaningful outcome and allow testing the stated hypotheses?

Reviewer #1: Yes

Reviewer #2: Partly

3. Is the methodology feasible and described in sufficient detail to allow the work to be replicable?

Reviewer #1: No

Reviewer #2: No

4. Have the authors described where all data underlying the findings will be made available when the study is complete?

Reviewer #1: No

Reviewer #2: Yes

5. Is the manuscript presented in an intelligible fashion and written in standard English?

Reviewer #1: Yes

Reviewer #2: Yes

You may also provide optional suggestions and comments to authors that they might find helpful in planning their study.

Reviewer #1: GENERAL COMMENTS

This study addresses an interesting and relevant topic. The introduction is well-written and provides a solid rationale. However, there are several areas that need clarification:

The study aims to: i) identify and characterize interventions used to treat FA in individuals with chronic musculoskeletal pain by considering all study designs, relevant measures, and FA synonyms; ii) evaluate the use of FA model-related measures in assessing treatment interventions; and iii) engage community collaborators in developing the systematic review methodology and reflecting on the findings to ensure clinical relevance.

The goals of this review align more closely with a scoping review rather than a systematic review.

The eligibility criteria should be framed using the PICOS (Population, Intervention, Comparator, Outcomes, Study Design) framework.

More detail is needed on the risk of bias assessment, data extraction, and the level of certainty evaluation.

Since the review does not aim to assess the effectiveness of interventions, the use of a certainty level assessment may not be appropriate.

INTRODUCTION

Line 121: The primary aim of the review should be to evaluate the effects of interventions for treating FA. A scoping review may be more suitable if the goal is simply to identify and characterize interventions.

METHODS

Line 174: The eligibility criteria should be organized according to the PICOS framework.

Line 175: As recommended by the Cochrane Handbook, no language restrictions should be applied. Additionally, the method for searching grey literature is not described.

Line 240: The inclusion and exclusion criteria should be placed under the "Eligibility Criteria" section. Please specify the criteria related to control groups and study design.

Line 260: Data extraction should be performed independently by two researchers.

Line 263: Data on the dosage of interventions (frequency, intensity, periodicity) should be collected.

Line 271: More detail is needed on the risk of bias assessment. Which domains will be evaluated, and how will the classifications "low risk of bias," "some concerns," and "high risk of bias" be determined?

Line 299: More information is required on how the GRADE system will be applied. What criteria will be used to assess the level of certainty, and how will high, moderate, low, or very low certainty be interpreted?

Line 299: It may not be feasible to assess the level of certainty using GRADE based on the data synthesis planned for this study. For example, when evaluating inconsistency, several criteria must be considered: i) the wide variance of point estimates across studies (note that direction of effect is not a criterion for inconsistency); ii) minimal or no overlap of confidence intervals (CIs), indicating variation greater than expected by chance; iii) statistical criteria such as heterogeneity tests, where a low p-value (p <0.05) suggests rejecting the null hypothesis; and iv) the I² statistic, which quantifies the proportion of variance due to between-study differences.

When assessing imprecision, one criterion is wide confidence intervals (i.e., the 95% CI includes no effect, and the upper or lower confidence limit crosses the minimal important difference [MID] for benefit or harm).

Since this study does not aim to assess the effectiveness of interventions, why is the level of certainty of intervention effects being evaluated?

Reviewer #2: Title

Lines 1-3. After reading the introduction and aims/objectives, I wonder if the title could be clearer. From the title, I thought this was going to be a review assessing the efficacy of interventions for FA and/or the epidemiology of fear avoidance in development of chronic MSK pain. I recognise that this isn’t what the title states and my inference is perhaps influenced by my own research priorities. But it may be worth considering if the title can be more clearly aligned to the aims/objectives so that prospective readers know exactly what this research is about.

Abstract

Well written and clear.

Strengths and limitations of this study

Lines 65-67. This point does not read well. The adjunctive word “however” makes it sound as though you are about to introduce an alternative perspective, but you instead make an additional supporting point for the same perspective.

Lines 65-67. You mention a scoping objective in this point, but at no point in the protocol do you elaborate on the scoping objective of this review.

Introduction

Introduced fear avoidance well with supporting literature including contemporary advancements of the term. Figure 1 is a good visual aid. You provide a good critique of the hyperfocus on kinesiophobia at the expense of other fear avoidance components.

Aims and objectives

Lines 121-127. The first objective is clear. The second objective – I am not certain what you intend to do; evaluate the FA related measures against what? And for what purpose? Do you just mean describe and characterise them, like the first objective? Or something else? I am also not sure what your third objective is – it seems vague. Considering this protocol is outlining your systematic review search methodology, is this objective not already achieved? And therefore, shouldn’t this be described within your methods rather than being an objective? Objectives should be geared towards what the results of your review will achieve. Please clearly state exactly what your 2nd and 3rd objectives are intending to achieve. These objectives may benefit from an overall superordinate aim.

Methods

The use of PRISMA-P is appropriate and all items are covered.

PPI

The description of PPI to inform the methods of this study are interesting and appropriate. The use of Twitter to reach clinicians in PPI is novel.

Eligibility criteria

Lines 175-178. This is very vague and will identify a huge number of citations and eligible full texts. You will end up excluding a number of these, the criteria for how you do this needs to be clear in your protocol – at the moment it seems like you will include them all? Please consider breaking this down into inclusion and exclusion criteria. Consider using PICOS/PECOS or a suitable alternative.

Please note that upon further reading I see that you have covered the above point elsewhere, but this is disjointed. All of the information should be together. Consider reorganising.

Search strategy

My initial feedback for this section included advice to include a finalised search strategy. Upon further reading, you indeed have included one. The reason for confusion is because of your use of the future tense for things you have already done – lines 187-189. This has cropped up in other places within the manuscript also – please ensure consistency and accuracy of past, present and future tenses. Please also consider the input of a librarian if possible to strengthen the reliability of your search strategy.

Line 218. Your “search 5” includes reference to an exclusion criteria (postoperative populations) – you haven’t detailed an exclusion criteria at this stage of your manuscript. As per previous point – consider reorganising.

Line 219. I would strongly advise against the use of line 40 in your search strategy. This will remove many potentially eligible papers from your search.

Lines 195-198 & lines 222-224. It is not necessary to provide such level of detail about how searching Ovid works as part of your manuscript – it’s not relevant to your study or normally detailed this much. Also, these functions change over time – so it’s likely these details may become incorrect at some point.

Study records

Lines 240-255. As per previous comments, consider moving this information to your eligibility criteria section. This is disjointed.

Lines 242 – do you have a definition or a reference point as to what constitutes a “musculoskeletal condition” vs what doesn’t? E.g., the ICD, or any generally accepted definition. Is carpal tunnel a musculoskeletal condition? How about peripheral neuropathy? Please expand.

Line 248 – I would reference the FA questionnaires to “Table 1” rather than “the above search strategy” – tables are not always in a linear order when published. Plus it’s more specific.

Line 258 – which researcher?

Confidence in cumulative evidence

Based on the last sentence here, line 299-300, it sounds to me as though you are confusing GRADE with a risk of bias assessment. GRADE does not lead to a quality based “study label”, it leads to a level of certainty in each individual finding you identify, informed by any number of studies (some of which may be high or low risk of bias, which makes up only one domain of GRADE). Additionally, GRADE is typically used with meta-analysis findings which you are not planning on doing. There is some guidance on how to apply GRADE for non-meta-analysis studies. Please provide details of what guidance you will use.

General feedback on Methods

You have not provided a reference for your systematic review methods. Whilst most of your methods are in keeping with best practice, some are not (e.g., one researcher performing data extraction). Please consider following a best practice guideline for systematic reviews (e.g., Cochrane, Joanna Briggs Institute, etc).

Am I right in thinking that you do not intend to make any inferences about efficacy of interventions used for FA? And that you do not intend make any inferences about the effectiveness of outcome measures in assessment of FA against a gold standard? This is an important point to be clear on since you are not planning on a meta-analysis. If you do intend to do either of those things, a meta-analysis should be considered.

Lines 293-294. Further to the above point, heterogeneity is not a well justified reason for not conducting a meta-analysis – you could use a random effects model and assess heterogeneity with an I2 statistic, and consider any heterogeneity in your interpretation of results. But it’s not clear to me what data you would be meta-analysing anyway based on your objectives. You mentioned in your strengths and limitations that a meta-analysis is not entirely applicable to the objectives – if this is true then your statement about heterogeneity is irrelevant as it’s not appropriate anyway. I would suggest further considering if a meta-analysis would be helpful or not – if it is, then plan for one (you may not end up doing it if the effect sizes are just completely incompatible); and if it is not helpful then remove the points about heterogeneity and instead explain clearly why it is inappropriate.

I am not familiar with your data synthesis methods. It would be helpful for me, as a reader, if you included more details of how these methods work specifically with analysing and synthesising data to answer your specific objectives. Essentially, I don’t fully understand what you are going to do. How are themes generated?

Discussion

Line 310 – this sounds more like a strength than a limitation to me?

Line 316-321. You didn’t provide details of how this will work in your methods. You identify it here as a potentially novel and therefore perhaps not yet defined method? I would outline this more clearly in the methods.

**Do you want your identity to be public for this peer review?** For information about this choice, including consent withdrawal, please see our Privacy Policy

Reviewer #1: No

Reviewer #2: **Yes: ** Michael Dunn

---

## [Author Response · Author response to Decision Letter 1]

23 Jan 2025

Dear Reviewers,

I hope you are well and thank you for taking the time to offer me a comprehensive overview of your comments. I have addressed every suggestion that you have noted below with an explanation beside each point. Please note that this paper still requires registration to a different protocol database, namely the Open Science Framework, as we have decided to change the study design to a scoping review to match its objectives, as you both kindly noted is more applicable. This is an ongoing process that we hope to be completed very soon. Otherwise, we hope the revisions made to be satisfactory.

A detailed response to all of your comments has been attached to the submission.

Many thanks,

Sam

---

## [Decision Letter · Decision Letter 1]

18 Feb 2025

Dear Dr. Watson,

Thank you for submitting your manuscript to PLOS ONE. After careful consideration, we feel that it has merit but does not fully meet PLOS ONE’s publication criteria as it currently stands. Therefore, we invite you to submit a revised version of the manuscript that addresses the points raised during the review process.

We look forward to receiving your revised manuscript.

Kind regards,

André Pontes-Silva

Academic Editor

PLOS ONE

Journal Requirements:

Reviewers' comments:

Reviewer's Responses to Questions

**Comments to the Author**

1. Does the manuscript provide a valid rationale for the proposed study, with clearly identified and justified research questions?

Reviewer #1: Yes

Reviewer #2: Yes

2. Is the protocol technically sound and planned in a manner that will lead to a meaningful outcome and allow testing the stated hypotheses?

Reviewer #1: Yes

Reviewer #2: Yes

3. Is the methodology feasible and described in sufficient detail to allow the work to be replicable?

Reviewer #1: Yes

Reviewer #2: Yes

4. Have the authors described where all data underlying the findings will be made available when the study is complete?

Reviewer #1: No

Reviewer #2: Yes

5. Is the manuscript presented in an intelligible fashion and written in standard English?

Reviewer #1: Yes

Reviewer #2: Yes

You may also provide optional suggestions and comments to authors that they might find helpful in planning their study.

Reviewer #1: The authors have adequately addressed all comments from the reviewers, and I have no further suggestions.

Reviewer #2: Abstract

Clear and well written

Strengths and limitations

Clear and well written. I am already clearer on the objectives of the scoping review than I was last time, even at this early stage, and why a meta-analysis isn’t appropriate. A good start.

Introduction

Clear and well written.

Aims/objectives

I am now much clearer on what the objectives are.

Methods

You now clearly state that you are using JBI guidance which is appropriate. You have also clarified your risk of bias assessment and removed GRADE appropriately, and addressed all other concerns except for the below two:

Eligibility criteria

This is much better organised. I can see that you have further expanded upon your definition of what constitutes musculoskeletal condition here, even differentiating between peripheral neuropathy and compressive neuropathy. But this definition is still vague and suggests that you will be interpreting what is and isn’t a musculoskeletal condition as you screen studies which isn't appropriate, rather than operating from a predetermined definition. I would still suggest that you provide a baseline definition of what constitutes a musculoskeletal condition e.g., injury/pain arising from bone, joints, ligaments, muscle, tendons, nerves – from memory I think that’s what the ICD classifies as a musculoskeletal condition. Another thing to consider is at what point a nerve injury becomes a neurological condition rather than an MSK condition and may therefore be excluded. You will find this crop up, for example, in whiplash associated disorder studies (which is an extremely relevant condition for fear avoidance) where grades 3 and 4 include nerve injury which may include people with neuropraxia or axonotmesis. Do you include these? If not then put this as part of the exclusion criteria. Other considerations include autoimmune MSK disorders and pelvic health MSK disorders, because these blur the lines between MSK and immune/visceral. If you don’t think this will impact your study objectives then perhaps not needed in your exclusion criteria. But worth considering if you haven’t already.

Search strategy

Noted that you still intend to use “NOT” as part of the search strategy. You have stated the nature of scoping review searching is iterative in line with JBI guidance as a justification for including “NOT” – this does not appear to be in the spirit of the guidance which details that the iterative nature pertains to “broadening the search as the researcher becomes more familiar with the evidence base, keywords and sources”. Noted within your manuscript, on this topic, you stated “This may mean incorporating those that can be discovered or removing ones that will decrease the sensitivity” (the one that decreases sensitivity being the “NOT”) which is contradictory to the excerpt of the JBI guidance I quoted. If I missed the bit of guidance you are referring to then please do direct me to the supporting part of the guidance. But otherwise, JBI guidance for scoping reviews states “Any limitations in terms of the breadth and comprehensiveness of the search strategy should be detailed and justified”. In line with this, please provide details for restricting the breadth of the search with “NOT” which may include things such as constraints of time and resources. Or if it is within the scope of your time and resources, please reconsider removing line 40 of the search in order to align with best practice for search strategies.

**Do you want your identity to be public for this peer review?** For information about this choice, including consent withdrawal, please see our Privacy Policy

Reviewer #1: No

Reviewer #2: **Yes: ** Michael Dunn

---

## [Author Response · Author response to Decision Letter 2]

1 Apr 2025

Reviewer #1: The authors have adequately addressed all comments from the reviewers, and I have no further suggestions.

Thank you for taking the time to review the revised manuscript.

Reviewer #2:

Abstract

Clear and well written

Strengths and limitations

Clear and well written. I am already clearer on the objectives of the scoping review than I was last time, even at this early stage, and why a meta-analysis isn’t appropriate. A good start.

Introduction

Clear and well written.

Aims/objectives

I am now much clearer on what the objectives are.

Methods

You now clearly state that you are using JBI guidance which is appropriate. You have also clarified your risk of bias assessment and removed GRADE appropriately, and addressed all other concerns except for the below two:

Eligibility criteria

This is much better organised. I can see that you have further expanded upon your definition of what constitutes musculoskeletal condition here, even differentiating between peripheral neuropathy and compressive neuropathy. But this definition is still vague and suggests that you will be interpreting what is and isn’t a musculoskeletal condition as you screen studies which isn't appropriate, rather than operating from a predetermined definition. I would still suggest that you provide a baseline definition of what constitutes a musculoskeletal condition e.g., injury/pain arising from bone, joints, ligaments, muscle, tendons, nerves – from memory I think that’s what the ICD classifies as a musculoskeletal condition. Another thing to consider is at what point a nerve injury becomes a neurological condition rather than an MSK condition and may therefore be excluded. You will find this crop up, for example, in whiplash associated disorder studies (which is an extremely relevant condition for fear avoidance) where grades 3 and 4 include nerve injury which may include people with neuropraxia or axonotmesis. Do you include these? If not then put this as part of the exclusion criteria. Other considerations include autoimmune MSK disorders and pelvic health MSK disorders, because these blur the lines between MSK and immune/visceral. If you don’t think this will impact your study objectives then perhaps not needed in your exclusion criteria. But worth considering if you haven’t already.

Lines 188-203/227-230: Thank you for your suggestions. I have provided the ICD-11 codes defining chronic musculoskeletal pain, both primary and secondary, and specified the exclusions. Chronic musculoskeletal pain secondary to nervous or visceral conditions will be excluded. However, autoimmune conditions will be included under chronic musculoskeletal pain secondary to persistent inflammation, as I stated my intention to include chronic secondary musculoskeletal pain (just not ones secondary to nervous/visceral).

Regarding whiplash-associated injuries, classified as chronic post-traumatic, these will be included. I have emphasized the necessity of a musculoskeletal component, and since grades 3 and 4 involve nerve injury, they still fall under the included code MG30.20 (chronic post-traumatic pain) and so will be included. Conversely, post-traumatic pain from injuries like burns has been excluded due to the absence of a significant musculoskeletal component.

Search strategy

Noted that you still intend to use “NOT” as part of the search strategy. You have stated the nature of scoping review searching is iterative in line with JBI guidance as a justification for including “NOT” – this does not appear to be in the spirit of the guidance which details that the iterative nature pertains to “broadening the search as the researcher becomes more familiar with the evidence base, keywords and sources”. Noted within your manuscript, on this topic, you stated “This may mean incorporating those that can be discovered or removing ones that will decrease the sensitivity” (the one that decreases sensitivity being the “NOT”) which is contradictory to the excerpt of the JBI guidance I quoted. If I missed the bit of guidance you are referring to then please do direct me to the supporting part of the guidance. But otherwise, JBI guidance for scoping reviews states “Any limitations in terms of the breadth and comprehensiveness of the search strategy should be detailed and justified”. In line with this, please provide details for restricting the breadth of the search with “NOT” which may include things such as constraints of time and resources. Or if it is within the scope of your time and resources, please reconsider removing line 40 of the search in order to align with best practice for search strategies.

Lines 249-251/276: I have removed all “NOT” as part of the search strategy as suggested as to not restrict the breadth of the search. Thank you for the clear explanation as I now realise the function “NOT” can have a significant restrictive impact to search results.

---

## [Editor Report · Decision Letter 2]

4 May 2025

An overview of the treatment interventions and assessment of fear-avoidance for chronic musculoskeletal pain in adults: a scoping review protocol

PONE-D-24-38361R2

Dear Dr. Fraje CE Watson,

We’re pleased to inform you that your manuscript has been judged scientifically suitable for publication and will be formally accepted for publication once it meets all outstanding technical requirements.

Kind regards,

André Pontes-Silva

Academic Editor

PLOS ONE
---

## [Editor Report · Acceptance letter]

PONE-D-24-38361R2

PLOS ONE

Dear Dr. Watson,

I'm pleased to inform you that your manuscript has been deemed suitable for publication in PLOS ONE. Congratulations! Your manuscript is now being handed over to our production team.

Kind regards,

on behalf of

Professor André Pontes-Silva

Academic Editor

PLOS ONE